# Effects of Transcutaneous Electroacupuncture Stimulation (TEAS) on Eyeblink, EEG, and Heart Rate Variability (HRV): A Non-Parametric Statistical Study Investigating the Potential of TEAS to Modulate Physiological Markers

**DOI:** 10.3390/s25144468

**Published:** 2025-07-18

**Authors:** David Mayor, Tony Steffert, Paul Steinfath, Tim Watson, Neil Spencer, Duncan Banks

**Affiliations:** 1School of Health, Medicine and Life Sciences, University of Hertfordshire, Hatfield AL10 9AB, UK; davidmayor@welwynacupuncture.co.uk (D.M.); proftimwatson@gmail.com (T.W.); 2MindSpire, Napier House, 14–16 Mount Ephraim Rd., Tunbridge Wells TN1 1EE, UK; tony@qeeg.co.uk; 3Max Planck Institute for Human Cognitive and Brain Sciences, P.O. Box 500355, D-04303 Leipzig, Germany; 4Statistical Services and Consultancy Unit, University of Hertfordshire, Hatfield AL10 9AB, UK; n.h.spencer@herts.ac.uk; 5School of Life, Health and Chemical Sciences, STEM, Walton Hall, The Open University, Milton Keynes MK7 6AA, UK; 6Department of Physiology, Busitema University, Mbale P.O. Box 1966, Uganda

**Keywords:** transcutaneous electroacupuncture, TEAS, EEG, HRV, eyeblink parameters, autonomic modulation

## Abstract

**Highlights:**

**What are the main findings?**
Eyeblink indices are potentially useful biomarkers for the effects of electroacupuncture and other neuromodulatory interventions.Significant autonomic correlates for some eyeblink parameters and EEG measures appear likely.

**What is the implication of the main finding?**
Traditionally viewed artefacts, such as eyeblink measures, may themselves possess diagnostic and research value.Transcutaneous electroacupuncture stimulation can modulate brain activity and autonomic function in a time- and frequency-dependent manner.

**Abstract:**

This study investigates the effects of transcutaneous electroacupuncture stimulation (TEAS) on eyeblink rate, EEG, and heart rate variability (HRV), emphasising whether eyeblink data—often dismissed as artefacts—can serve as useful physiological markers. Sixty-six participants underwent four TEAS sessions with different stimulation frequencies (2.5, 10, 80, and 160 pps, with 160 pps as a low-amplitude sham). EEG, ECG, PPG, and respiration data were recorded before, during, and after stimulation. Using non-parametric statistical analyses, including Friedman’s test, Wilcoxon, Conover–Iman, and bootstrapping, the study found significant changes across eyeblink, EEG, and HRV measures. Eyeblink laterality, particularly at 2.5 and 10 pps, showed strong frequency-specific effects. EEG power asymmetry and spectral centroids were associated with HRV indices, and 2.5 pps stimulation produced the strongest parasympathetic HRV response. Blink rate correlated with increased sympathetic and decreased parasympathetic activity. Baseline HRV measures, such as lower heart rate, predicted participant dropout. Eyeblinks were analysed using BLINKER software (v. 1.1.0), and additional complexity and entropy (‘CEPS-BLINKER’) metrics were derived. These measures were more predictive of adverse reactions than EEG-derived indices. Overall, TEAS modulates multiple physiological markers in a frequency-specific manner. Eyeblink characteristics, especially laterality, may offer valuable insights into autonomic function and TEAS efficacy in neuromodulation research.

## 1. Introduction

Transcutaneous electroacupuncture or electrical ‘acupoint’ stimulation (TEAS) is used increasingly in clinical practice as a non-invasive alternative to electroacupuncture using needles. Most common applications are for the alleviation of pain, nausea and vomiting, postoperative recovery, and in vitro fertilisation [1].

Eye movements such as blinking are usually considered as contaminating artefacts in electroencephalography (EEG) research that should be removed to enable better analysis of brain function. However, eyeblink characteristics may also be useful indicators in themselves. Blink rate (frequency), for example, typically between 10 and 25 blinks per minute at rest, may increase with fatigue, stress, anxiety, or nervousness, but may decrease in Parkinson’s disease or during tasks that require intense concentration or cognitive effort [2]. Other eyeblink parameters are provided as output from the easy-to-use MATLAB^®^ software package BLINKER [3], including several variants of duration, peak maximal values, and ‘amplitude–velocity ratios’ [4]. These are listed in the accompanying online Appendix A and described in more detail on the GitHub BLINKER page [5]. Versions R2024a and 1.1.0 were used for MATLAB^®^ and BLINKER, respectively.

A novel index that we introduce here is ‘best’ blink laterality, referring to the tendency for blinks to occur more frequently in one eye compared to the other (detailed explanation below, based on Kleifges et al. (2017) [3]). We found little previously published material on eyeblink laterality prior to BLINKER, other than a study indicating that eyelid (blink) asymmetries decrease with increasing experimental trigeminal stimulation but are in general stable over several months [6], together with several studies on ‘pre-pulse inhibition’ (PPI) of the startle response [7]. In one of these, Braff et al. cited previous research in which blink reflex magnitude was found to be greater for the right than for the left eye in the general population [8]. PPI itself was also noted to be greater in the right eye in healthy individuals, and particularly in smokers, but this difference was reduced in patients with schizophrenia [9]. Nicotine is well known to affect blink rate, presumably via a dopamine-mediated mechanism [10].

In 2013, aware that eyeblink rate (EBR) may be an indirect marker of striatal dopamine activity [10,11,12] and inversely correlated with parasympathetic activity [13,14], we presented a preliminary report based on four small pilot studies (*N* participants = 1 to 12) [15]. Our findings suggested that EBR increased more with electroacupuncture (EA), i.e., electrical stimulation applied using needle electrodes, than with gentle manual acupuncture (MA), that EBR increased more with 20 min than 5 min of EA, and that EBR decreased following EA. These results were all statistically encouraging, with *p*-values < 0.001 for Wilcoxon signed-rank and binomial tests. A finding that did not attain statistical significance was that EBR was usually greater for transcutaneous EA stimulation (TEAS)—i.e., transcutaneous electrical nerve stimulation (TENS) applied to acupuncture points—than for MA. In addition, we observed differences in EBR that failed to reach statistical significance with stimulation frequency in cycles per second (2.5 Hz or 10 Hz), and that EBR appeared to increase more with 20 min than with 5 min of EA.

Here, we analyse eyeblink data from a larger and more recent study on TEAS alone (2015–2016, *N* = 66, of whom 41 were women and 25 were men), comparing the effects of different stimulation frequencies in four sessions spaced a week or more apart. We also extended the range of indices provided by BLINKER by using their time series as input for CEPS, an open access MATLAB**^®^** software package for the analysis of complexity and entropy in physiological signals [16,17]. We then compared effect sizes for the effects of TEAS frequency, session order, and time within sessions on the original and extended BLINKER measures, other EEG-derived indices, and different measures of heart rate variability (HRV) and pulse rate variability (PRV).

On the supposition that there might be a central ‘frequency following effect’ in response to peripheral rhythmical stimulation, such as TEAS [18], with low-frequency stimulation enhancing low-frequency (delta, theta) EEG power and so tending to relax, and high-frequency stimulation increasing high-frequency (beta, gamma) EEG power and so tending to stimulate [19], our original study objective was to explore the effects of different frequencies of TEAS, rather than of stimulation amplitude [20]. Since then, at least one research group has explored this possibility, even writing of ‘theta and gamma electroacupuncture’ (stimulation at 6 Hz and 40 Hz, respectively) [21,22]. Another group noted that low-frequency (but not high-frequency) median nerve stimulation (TENS) enhanced occipital EEG alpha power spectral density (PSD) and reduced frontal Beta1 PSD in patients with generalised anxiety disorder [23]. In contrast, other researchers found that 100 Hz TENS applied locally to the forearm at either side of a pressure cuff used to induce ischaemic pain *reduced* the increase in gamma power that occurred when the cuff was inflated [24].

In 2021, we analysed the data from our 66 study participants in a conference presentation extending the range of autonomic measures associated with heart rate variability, in which we categorised the HRV measures output by the Kubios HRV software (Kubios Oy, Kuopio, Finland) as indicating either predominantly parasympathetic (‘PNS-like’) or sympathetic (‘SNS-like’) modulation, ‘ambivalent’ or ‘other’ effects. These same categories were also used to classify a variety of non-HRV measures derived from the electrocardiography (ECG), photoplethysmography (PPG), and respiration data collected concurrently [25]. Although it would be overly simplistic to consider the two main divisions of the autonomic nervous system, the parasympathetic and sympathetic, as functionally polar opposites [26], here, we extend this classification to a number of different EEG data types (see Appendix A). As an aside, although PPG data were collected, pulse rate variability (PRV) was not analysed in detail independently (although similar in some respects, HRV and PRV provide different information on underlying physiological function [27]).

Our present objectives are as follows:

Objective 1. To consider whether time, stimulation frequency, or session order has a greater effect on eyeblink, EEG, and HRV indices/measures, and whether the eyeblink indices offer any advantage over the EEG and HRV measures in this regard.

Objective 2. To explore some possible parasympathetic/sympathetic associations of these measures, comparing strengths of association for the different eyeblink and EEG-derived data types.

Objective 3. To investigate whether these measures may be predictive of participant dropout or adverse reactions to stimulation.

## 2. Materials and Methods

### 2.1. Data Collection

EEG, ECG, and other data were recorded in the University of Hertfordshire’s physiotherapy laboratory, as described in detail elsewhere [28,29,30]. The University Health and Human Sciences Ethics Committee granted ethical approval (protocol number HSK/SF/UH/00124). 

The study design was that participants should attend for four planned sessions, during each of which TEAS was applied to both hands over the acupuncture point *hegu* (LI4) and a point on the ulnar border of the same hand (so that current did not cross the midline of the body and did not flow through the arms and torso, and thus, in principle, should not affect the heart—or brain—directly). A different stimulation frequency was applied in each session—at 2.5, 10, 80, or 160 pulses per second (pps)—with the 160 pps frequency used as a ‘sham’ treatment. 

To avoid repeated order effects, the different frequencies were applied in semi-randomised order: the sequence of sessions was first randomised, then checked to ensure no particular frequency was preponderant, and the sequence was amended if this was the case. For the active treatments, amplitude was increased until participants described it as ‘strong but comfortable’ [31], while for the sham treatment, amplitude was very low, with the amplitude control on the charge-balanced biphasic square-wave Equinox E-T388 stimulator (Equinox, Liverpool, UK) used being turned to zero once it had been felt initially by the participants, but with a flashing indicator lamp still visible (a procedure similar to that used previously in both clinical and experimental studies of transcutaneous electrical nerve stimulation (TENS) [32,33,34] and TEAS itself [35,36,37]). Participants were informed that they would habituate to the sham stimulation and would be unlikely to feel it after a short time. Nonetheless, only 19 (30.6%) reported after their sham session that they had been unable to feel anything at all during it. 

On first arriving, the experimental procedure was explained to the study participants. As part of this, they were told: “We are not asking you to meditate or do anything out of the ordinary, but just to keep your eyes open and blinking as little as possible” (during the experiment, if participants were observed to blink rapidly, they were also asked by one of the researchers to slow their blinking down). Every attempt was made to keep the participants comfortable. Electrode caps were selected individually, in accordance with participant head size, to try and minimise any feelings of constriction or compression. Following the completion of some initial questionnaires and a numerical rating scale (NRS) for mood [29,38], the various electrodes and sensors were attached, including a 19-channel ECI (Electro-Cap International, Inc., Eaton, OH, USA) EEG cap (electrodes positioned according to the international 10/20 system, and with linked ears as reference and ground anterior to Fz), forearm ECG, fingertip photoplethysmography (PPG), finger temperature, respiration belt, and head movement sensors. Data were then recorded in eight consecutive 5 min time slots—baseline (Slot 1), stimulation (Slots 2–5), and post-stimulation (Slots 6–8). The NRS instrument was also completed following Slots 3 and 8. 

### 2.2. Participants

Sixty-six participants were recruited over a period of 14 months as a convenience sample from university staff and students, and from other interested local contacts (including several established complementary health practitioners). This sample was expected to be typical of the general population.

### 2.3. Data Recording and Preprocessing

EEG data were recorded at 500 Hz using a Mitsar EEG-202 amplifier, and WinEEG software (v2.91.54) (Mitsar Ltd., St. Petersburg, Russia). Preprocessing steps and artefact removal are described elsewhere [30]. 

ECG data were recorded using the same equipment and software, with standard ECG electrodes (Kendall™ H124SG, Cardinal Health UK, Leeds, UK) positioned bilaterally on the volar surfaces of the forearms. In addition, respiration and head movement data were recorded using, respectively, an abdominal piezo crystal SleepSense respiration effort sensor belt (Scientific Laboratory Products, Elgin, IL, USA) and a head-mounted Pico Micro Movement Sensor (Unimed Electrodes, Farnham, Surrey, UK); these data were not analysed for the purposes of the current paper. ECG artefact correction and detrending (using the ‘smoothness priors’ method) were conducted using Kubios HRV software (Kubios HRV Standard, 3.3.0, Kuopio, Finland). 

Finger temperature was recorded using a NeXus-10 physiological monitoring unit with its associated temperature sensor and BioTrace v2015B software (MindMedia BV, Herten, The Netherlands). The same unit and software were also used for recording bilateral PPG and a further channel of ECG (results not analysed here). Temperature was recorded at 32 Hz, recordings trimmed to remove obvious initial artefacts and then smoothed using the method available in the BioTrace software, but otherwise not preprocessed. 

### 2.4. Measures Analysed


1.BLINKER


BLINKER considers as potential blinks only those events that last more than 50 milliseconds (ms) and are at least 50 ms apart. A single EEG (or electrooculogram, EOG) channel is then selected by the algorithm to provide the ‘best’ blinks, defined by the closeness of their actual trajectory to that of a ‘stereotypical’ blink, with correlation between them (Pearson’s R^2^) ≥ 0.98; for ‘good’ blinks, R^2^ ≥ 0.90 [3]. Parameters of the ‘stereotypical blink’ were not defined by Kleifges et al. [3], as appears to be the case for other studies in the literature in which stereotypical blinks are simply manually selected without further discussion (e.g., [39,40]). However, Kay Robbins, the communicating author of the original BLINKER study, has clarified that by ‘stereotypical blink’, the authors meant the ‘normal’ blink described by Johns [4], namely one with a single maximum whose derivative also has a single maximum, but followed by a single minimum [41]. More generally, it has been suggested that “all blinks of a test subject look pretty much alike, even among independent sessions. While blinks of different persons seem to look slightly different, they appear to be still highly correlated. Hence, it is possible to use a single prototype blink with the signals of different subjects” [42]. Miyakoshi et al. [43] made the interesting observation that physical effort to suppress blinks may indeed prevent the generation of stereotypical blink-induced EOG waveforms. 

We considered several basic ‘families’ of measures output by BLINKER (for details, see Figure 1 here and Appendix A):(1)Duration: median, mean, and ‘mad’ (mean absolute deviation) blink durations (at blink base and zero levels, as well as at half-base and half-zero levels, together with ‘tent’ duration), with corresponding measures for ‘good’ blinks (goodMedian, GoodMean, etc.).(2)Blink rate (blinks per minute, BpM)—mean, or mean of ‘good’ blinks per minute—together with numbers of blinks and of ‘good’ blinks.(3)Amplitude–velocity ratios (nAVRB, nAVRT, and nAVRZ), estimated from intervals between peak maximum and right base, between tent peak and right tent line, and between peak maximum and right zero; correspondingly for pAVRB, pAVRT and pAVRZ and right blink markers.

In addition to this subset of 61 of the standard ocular indices provided by BLINKER, we also considered the ratio of left to right channels providing the best blinks (LRBR), as well as counts of left and right best blinks, resulting in 63 indices (see Appendix A). Furthermore, we fed the output of all the ocular indices from BLINKER for which this seemed appropriate to CEPS [16,17], to calculate a second layer defined as CEPS-BLINKER composite measures. Although less immediately interpretable in terms of physiology than the BLINKER indices, these would provide estimates of the complexity and variability/predictability of the ocular indices themselves. Given that the number of blinks in each 5 min recording was relatively small (see Section 3 below), only 34 of the CEPS measures could be used for this purpose, based on 30 of the original BLINKER indices, resulting in 34 × 30 or 1020 composite measures. Details are provided in Appendix A.

For comparison purposes, as a convenience sample, a number of EEG power-based data types were also used, selected partly because their analysis was possible using the skills and resources available to us at the time, and on the basis of prior research indicating their potential usefulness for this project. Brief descriptions follow.

2.Median power for each 5 min recording, for all 228 channel (19) and band (12) combinations.

The EEG bands used were the five standard bands (delta, theta, alpha, etc.), together with Beta1 (15.0–18.0 Hz), Beta2 (18.0–25.0 Hz), Beta3 (25.0–35.0 Hz), the sensorimotor rhythm (SMR, 12.0–15.0 Hz), 2.0–20.0 Hz and 0.5–45.0 Hz, with one other (‘Infill1’, 12.0–19.0 Hz). EEG power is a usual starting point used in investigations of the effects of different interventions on the central nervous system. Prior research on EA frequency and EEG power can be found in the literature (e.g., [45,46]). As mentioned, the correlations between EEG features and HRV or stress measures have been researched by a number of authors [47,48,49,50,51]. 

3.Regional power ratios: posterior/anterior, left/right, central/outer ratios of absolute or relative power in 12 standard and ‘infill’ EEG bands—a total of 252 measures.

We initially considered these regional ratios as a means of differentiating between volume (outer) and neural (central/somatosensory) conduction effects [52] and then added the other regional ratios as well [53]. Changes in EEG laterality have been reported before in response to TEAS [54] and TENS [55]. To our knowledge, the posterior/anterior (P/A) ratio has not been used in acupuncture-related research before, and although EEG laterality may be affected by stress [56], the P/A and central/outer (C/O) ratios have not, to our knowledge, been considered in relation to HRV. 

4.Eighteen median power *asymmetries*, for both symmetrical right/left electrodes and regions [Log_e_(Mdn Pwr Right) − Log_e_(Mdn Pwr Left)], and for posterior/anterior and peripheral/central regions, in the same bands as for median power, together with 6 narrow bands centred on frequencies of potential interest (stimulation frequencies or their sub-multiples) and a further 12 narrow bands centred on frequencies presumed unrelated to stimulation frequencies, with 18 absolute and relative powers in narrow bands—a total of 602 measures.

Frontal alpha band asymmetry is a frequently used measure in EEG research [57], with changes reported in response to acupuncture, for example [58]. In particular, frontal alpha asymmetry is commonly used in studies on stress [48,51]. Here, we assessed EEG asymmetries more generally, not just in the alpha band. 

5.Median power asymmetry *ratios* [e.g., 2 × (Log_e_(Mdn Pwr Right) − Log_e_(Mdn Pwr Left))/(Log_e_(Mdn Pwr Right) + Log_e_(Mdn Pwr Left))], as for the asymmetries but omitting narrow bands centred on 120 Hz and the absolute and relative power bands—a total of 530 measures. We also considered power asymmetry *ratios* [59] as variant measures of asymmetry.6.Cordance measures in various standard and non-standard EEG bands, for all electrodes, both non-normalised and calculated as a comparison using both log-normalised [60,61] and square-root-normalised algorithms [62], with bands created using Thomson multi-taper or continuous Morlet wavelet methods [63,64]—a total of 8148 measures.

Leuchter and colleagues have claimed that cordance, a measure they derived from averaged absolute and relative EEG power in a particular band measured from all or some of the electrode pairs used [65], “is potentially a more specific indicator of cerebral dysfunction than QEEG [i.e., quantitative EEG] power, since it has normal and abnormal indicator states” [66]. They also noted that greater EEG *alpha* represents the inhibition of brain activity, thereby reflecting a reduction in brain perfusion in the healthy [62]. Given that the frequency of TEAS or EA may affect circulation (e.g., [67]), we considered that cordance would, therefore, be a useful measure to use in the present study. To our knowledge, no studies on interactions between HRV and cordance have been published. One study on cordance and ‘dry needling’ (a modern, stripped-down version of traditional acupuncture) was located [68], but none on cordance and EA or TENS/TEAS. 

7.Centroids: regional power and frequency centroids; electrode spectral centroids; finger temperature slope (over time). ‘x’ and ‘y’ coordinates of centroids of all electrodes, those on the left or right, anterior or posterior, central or peripheral regions of the cranium, together with spectral centroids for each of the 19 channels—a total of 48 measures.

Sulaiman et al. (2010) [69] used EEG spectral centroids for identifying stress, and Giannakakis et al. (2015) [70] found that EEG centroid frequencies in many channels did indeed appear to increase with stress. We therefore considered this method appropriate here and also used spatial centroids as another method of identifying particular brain regions that might be affected by different frequencies of TEAS.

Acute stress may trigger peripheral vasoconstriction, causing a rapid drop in skin temperature [71]. Skin temperature itself may increase or decrease in response to EA, depending on the individual [72]. Finger temperature slope was investigated here to explore the autonomic effects of different frequencies of TEAS and whether these were experienced as stressful or not. Low-frequency stimulation was expected to result in vasodilation rather than vasoconstriction [67].

8.Power in 14 1 Hz bins, either bands centred on frequencies of potential interest (stimulation frequencies or their sub-multiples) or centred on frequencies presumed unrelated to stimulation frequencies, with bands created using the methods mentioned above. Two different methods of independent component analysis (ICA) were used in the data preparation: Extended InfoMax [73] (1065 measures) and Adaptive Mixture ICA, or AMICA [74] (1063 measures). Preprocessing was either using the standard methods available in EEGLab v 2022.0 [75] and associated software, using a pipeline created by Paul Steinfath as described elsewhere [30], or a semi-automated machine learning method developed by Thea Radüntz (who kindly provided the results for us) [76]—a total of 2128 measures.

Our initial hypothesis [18] was that peripheral stimulation as applied in this study might induce a ‘frequency-following response’ (FFR) centrally, in the EEG. Using narrow frequency bins, we hoped to detect any resulting FFR. Given that correlations have been found between the HRV HFnu index and EEG Alpha2 (a narrow band between 10 and 12 Hz) [49], we also hoped to detect such correlations in our own data. 

9.In addition, the ‘classical’ EEG Hjorth activity, complexity and mobility parameters [77,78] were computed for each electrode (228 measures), as well as Wackermann’s ‘global descriptor’ indices sigma, phi and omega [79] (median values at the start, middle, and end of each recording) (48 measures).10.A further dataset consisted of the Wackermann descriptors for the 4 sec segments in each 5 min slot, with a repeat of the median values from the previous dataset, based on either Infomax or AMICA ICA (3188 measures).11.Finally, HRV and PRV (pulse rate variability) indices computed using Kubios HRV software (Standard version 3.3.0) were included, for both the 5 min recordings (59 PRV and 59 HRV measures) and their 1 min segments (only a further 66, i.e., 2 × 33, measures, as 1 min segments were too short to compute some measures).

To our knowledge, neither the Hjorth parameters nor Wackerman’s global descriptors have been used in prior acupuncture research, nor have they been researched for their association with HRV measures. However, they do provide an overview of the EEG that we considered could be useful in this study. 

In total, some 15,356 measures were available for analysis. 

Data for all these measures will be made available by the corresponding author (DB, email duncan.banks@open.ac.uk), upon reasonable request. Some findings for the most salient measures are presented in Table 7 below, and in Appendix A.

### 2.5. Statistical Analysis

Analysis was conducted in various stages.


(1)Slot Values:


Initially, 5 min values of the various measures were analysed, or their median values for each 5 min recording when more appropriate.


(2)Slot Differences (values normalised with respect to baseline, within session):


To take into account differences at baseline (in Slot 1) that could clearly not be attributed to the effects of stimulation frequency, the authors decided to also examine normalised values, or differences between values in a given slot and in Slot 1.

Shapiro–Wilk tests were used (in both SPSS and *R*) for both values and differences to determine whether data were normally distributed or not. Preliminary tests for normality of distribution were initially also carried out in Excel using the Jarque–Bera test [80].


12.Given the large number of variables involved in this exploratory study, most would be expected to provide little, if any, useful information on any questions we might ask of the data. In an earlier project, we explored the effects of space and terrestrial weather conditions on EEG and ECG readings collected during TEAS sessions, investigating more than 7700 features using a method called highly comparative time series analysis [81]. Here, we used a different approach, making use of what we call ‘top slicing’ methods in order to reduce the number of measures considered to something more informative and manageable. Various methods of ‘top slicing’ were compared to focus on those measures most likely to show significant differences with stimulation frequency, session, or slot within session. These are discussed below, in the Results section. Friedman’s test for repeated measures was used for both values and differences (as a non-parametric equivalent of analysis of variance, or ANOVA), with Kendall’s *W* as a measure of effect size, and Wilcoxon and Conover–Iman matched-pair signed-rank tests for post hoc analysis.


(3)Bootstrapping (random sampling with replacement):

To provide more robust results for the Friedman, Kendall, Wilcoxon, and Conover–Iman tests, ‘bootstrapping’ of 1000 randomly sampled sets of results was used [82], with results reported as medians and 95% confidence intervals. 

(4)Interactions between frequency, session, and slot within session were also tested for, using the simple rank test for interaction proposed by Hettmansperger and Elmore (2002) [83].(5)Other non-parametric tests were used as required, such as the Mann–Whitney *U* test (also known as the Wilcoxon rank-sum test) for differences between independent samples and Spearman’s rank correlation method for correlations between them. For the latter, confidence intervals were calculated using the method of Bonett and Wright [84].


13.Effect size: In addition to statistical significance (using *p*-values), effect size (ES) was computed using different methods, as appropriate for each test. For Mann–Whitney tests, Cohen’s *r* (or *Z*/√*n*) was used, where *Z* is the *Z*-score, and *n* is the total number of observations on which *Z* is based [85,86]; for Friedman and Conover tests, Kendall’s *W* (or coefficient of concordance) was used, linearly related to Friedman’s *χ*^2^ (*W* = *χ*^2^/*n*(*k* − 1), where *k* is the number of measurements per participant [86]. For Spearman’s rank correlations, the correlation coefficient itself (*rho*, *ρ*) was taken as a measure of ES [87]. For all three methods, the usual convention was adopted: 0.1 ≤ ES < 0.3 was considered ‘small’, 0.3 ≤ ES < 0.5 as ‘medium, and ≥ 0.5 as ‘large’.


The following software packages were used: IBM SPSS Statistics v. 28 (IBM, Armonk, NY, USA); RStudio 2024.04.0+735 “Chocolate Cosmos” Release for Windows (RStudio Team, 2020); and RStudio: Integrated Development for R. RStudio (PBC, Boston, MA, USA), with subsidiary packages for the Friedman, Kendall, Wilcoxon, and post hoc Conover tests (the latter amended by author NS).

When discussing statistical significance, the usual 5% level of significance was employed, except when otherwise specified, but we must be mindful of the fact that many comparisons can be carried out. In these circumstances, *p*-values should not be taken at face value and we have two options: (i) make adjustments to the *p*-values so that they can be interpreted in a traditional manner; (ii) treat the *p*-values merely as indications of the strength of evidence against the null hypothesis without any more formal interpretation. In the publication of most studies, option (i) would be favoured over option (ii). However, traditional methods for *p*-value adjustment (e.g., Bonferroni, Šidák) are only appropriate for situations where the multiple tests are independent of each other, and this is far from being the case here. For situations, such as the case in this paper, where the multiple tests are not independent, simulation methods can, in theory, be used to make adjustments (e.g., [88]). However, in the current context, with a very large number of highly correlated tests, it was not computationally feasible to undertake such simulations. As a result, option (i) for dealing with the multiple comparisons issue was not available, and we were obliged to use option (ii), taking the *p*-values as they were and applying appropriate scientific judgment and caution when drawing conclusions.

## 3. Results

### 3.1. Participant Demographics

Of the 66 participants recruited, 4 dropped out after only one session, and another only completed three sessions. Technical and other issues (such as a computer crash on one occasion and a participant ignoring the instruction to visit the toilet before a session, so that it had to be interrupted) led to incomplete data in a number of other cases. Complete datasets were obtained for 48 participants (18 men and 30 women, aged between 18 and 69). Their ages were not evenly distributed, and although more women (F) than men (M) took part in the study, this was not the case in all age groups, as shown in Appendix A.

The flowchart in Figure 2 summarises the above, from recruitment to statistical tests used.

### 3.2. Normality of Distribution

Testing for the normality of the distribution of measure values using Shapiro–Wilk tests indicated that BLINKER, CEPS-BLINKER, and most of the other data types used were not normally distributed, except for some EEG Median power, median power asymmetry and asymmetry ratio results, justifying the use of non-parametric rather than parametric statistical tests for the remainder of the study. 

### 3.3. Top Slicing Methods Used

We considered taking only those in the upper quartile, the top 10%, top 5%, top 1% or top 0.1% of values or counts of Friedman’s chi-square, Kendall’s *W* or the Conover–Iman statistic (CIS), counts of these three statistics greater than a certain threshold (e.g., 1, 2 or 3 for the Conover–Iman statistic), or simply the top 10 or top 2 measures—or even the single very top measure—for each data type. For example, differences in BLINKER measures between slots for each stimulation frequency (4 × 61 rows in the *R* output file) resulted in Spearman correlations between the results from the different top slicing methods with *rho* > 0.8 and *p* < 0.001 regardless of whether strengths of the correlations between the resulting 45 pairs of percentiles, counts and/or Kendall’s *W* were checked. Thus, in a way, it almost seems irrelevant which method of top slicing should be selected, whether these are based on percentages or counts greater than a certain threshold. We eventually settled on selecting the ‘top 10’ measures for each data type as a pragmatic way of proceeding. However, different methods could be used to select the ‘top 10’ measures for each data type, such as either *including* or *excluding* repetitions. In the former, the same measure may appear in several rows (e.g., for several frequencies when testing for differences in measures between slots for each stimulation frequency); in the latter, only for a single row in the *R* output. As an example, the ‘top 10’ BLINKER measures using each method when testing for differences in measures between slots for each stimulation frequency are shown in Appendix A, with six measures occurring in both lists. 

**Results for Objective 1.** Does time, stimulation frequency, or session order have a greater effect on eyeblink, EEG, and HRV indices/measures, and do the eyeblink indices offer any advantage over the EEG and HRV measures in this regard? 

Counts of the ‘top 10’ measures in the results for Friedman’s chi-square or for Kendall’s *W* will be very much the same, given the linear relationship between the two statistics. The results for Kendall’s *W* are shown in graphical form in Figure 3, for all the data types shown in the study flowchart (Figure 2) taken together. For example, counts of differences between slots varied with stimulation frequency, being greatest at 2.5 pps (70+ out of 140 measures) and least for sham (only around 10 of 140 measures) (Figure 3A). Counts of differences between slots were also greatest in Session 1 (Figure 3B), whereas counts of differences between TEAS frequencies were greatest during stimulation (Slots 2–5) (Figure 3C). Note that counts of the ‘top 10’ measures are not the same as those of the 10-scoring measures, which will always be equal to 10. 

Those of the ‘top 10’ measures that occurred three times or more for both Values and Differences—i.e., for three or more comparisons (4 × Stimulation frequency within Slot, 8 × Slot within Session, etc., including 32 Slot × Stimulation frequency combinations for Values, but not for Differences)—are shown in Appendix A. 

Note that for 19 of the ‘top 10’ measures for all data types in Appendix A, the effect size (Kendall’s *W*) was greater for differences, whereas this was only the case for 11 values. Considering only the single top-most measure for each data type, Kendall’s *W* for differences between stimulation frequencies within each slot was now more often greater for values (nine data types) than for differences (six data types). Corresponding results for Friedman’s *Chi*^2^ are shown in Figure 4. Note, too, that nearly all the effect sizes (median *W*) were small (0.1 ≤ ES < 0.3), if not very small. Incidentally, for *all* of the top 10 measures for ‘1 Hz bins’, *W* for both values and differences was >0.2 for differences between slots for each stimulation frequency, and also for differences between slots for each session, but not for the other value or difference comparisons. Maximum effect sizes were found for the 1 Hz bins, followed by median power (values: O2.5_P3, *W*_max_ = 0.3023; differences: O5_P3, *W*_max_ = 0.3258). These were for differences between stimulation frequencies within each slot, not differences between sessions within each slot. Corresponding results for Friedman’s *Chi*^2^ are shown in Figure 4. The results for the 1 Hz bins are provided in more detail in Appendix A. 

More graphical results comparing values and differences (values normalised with respect to baseline within session) are included in the Appendix A. 

### 3.4. Comparing Maximum Effect Sizes for Differences Between Sessions and Differences Between Stimulation Frequencies, Both Within Slots

Figure 5 shows the maximum effect sizes for differences between sessions and differences between stimulation frequencies within slots. 

Median *W* over all data types: Session by slot (Diffs): 0.129;Stimulation frequency by slot (Diffs): 0.179;Session by slot (values): 0.126;Stimulation frequency by slot (values): 0.127.

Overall, ES is marginally greater for differences than for values, and, more obviously, for differences between frequencies than for differences between sessions. 

The lowest values of *W* are consistently for the regional ratios, followed by centroids, highest values for median power in 1 Hz bins and in the standard EEG bands. BLINKER and CEPS-BLINKER measures do not perform particularly well in differentiating between stimulation frequency and session. As might be predicted, finger temperature and HRV/PRV do not show a session order effect. 

### 3.5. Differences over Time (Between Slots) Within Sessions

For the majority of data types considered (8 out of 11), the 99th percentile of Kendall’s *W* is greatest for changes over time (i.e., when comparing time slots *within* sessions) than when comparing results for the four different sessions or the four stimulation frequencies, as can be seen in Appendix A. 

From Appendix A, it is clear that Kendall’s *W*, as a measure of effect size, was most commonly very small (<0.1), although >0.1 for five data types (including BLINKER) for time slot comparisons, and for the frequency and session comparisons for cordance only. 

### 3.6. Post hoc Results, Using the Conover–Iman Statistic (CIS)

Turning now to the post hoc results, it is clear that findings for the Conover–Iman statistic follow a similar pattern to those for Kendall’s *W* (Appendix A). Furthermore, counts of the Conover–Iman statistic (CIS) > 3 (as well as of the 95th percentiles of the CIS) indicate that differences from baseline (Slot 1) increase with time, when all 63 BLINKER measures are considered together (Figure 6). Early changes (between Slots 1 and 3) appear to be greater for 80 pps stimulation, with a cumulative effect for all stimulation frequencies by Slot 5, although this is most consistently maintained only for 10 pps stimulation thereafter. 

Counts (and values) of the blink measures appear to increase more rapidly during 80 pps than 2.5 pps stimulation, with more of a consistently cumulative effect over time for 10 pps. The apparent pattern of increases only for every other slot is in part due to the cumulative effects of stimulation (in Slots 2–5) and a subsequent decrease post-stimulation (in Slot 6), together with a steady increase in CIS over time. An overview of CIS results for the BLINKER and CEPS-BLINKER measures that differentiate between slots at each stimulation frequency, between frequencies, and between sessions within each slot is provided in Table 1. 

From this top-level analysis, changes over time *within* sessions—whether for the BLINKER measures or for the CEPS measures derived from them—are greater than their differences with stimulation frequency, and greater than their differences over time between sessions. However, contrary to expectation, differences in the CEPS measures derived from the BLINKER indices are less over time (within sessions) than those for the BLINKER measures themselves. Nonetheless, the *maximal* values of the CEPS-BLINKER measures are indeed more than the maximal BLINKER measures themselves when comparing their values for the different stimulation frequencies, as well as for the different sessions. 

### 3.7. Measures and Indices Most Affected by Stimulation Frequency, Time Slot, or Session Order

The ten CEPS measures and BLINKER data types from which they were derived that provided the greatest differentiation between stimulation frequencies, between times, and between sessions (the largest 95th percentiles of CIS) are shown in the Appendix A.

When *all* data types are considered (1 Hz bins, HRV, etc.), the patterns of change from baseline are similar to those for the 63 BLINKER output measures shown in Figure 6 above (also see Appendix A)—namely, consistent increases from Slots 1, 2, 3 and 4 (before and during stimulation), with subsequent decreases from Slots 5, 6 and 7 (post-stimulation). The top *two* measures most affected, together with the Frequency, slot or session pair for which they showed maximum values of the Conover–Iman statistic, are listed in Appendix A. Note that pairs are rarely consistent for the two measures. The most commonly occurring pairs, i.e., greatest differences, were time ‘Slots’ 2–7 (four occurrences), Sessions 1–4 (eight occurrences), and stimulation frequencies 2.5–80 pps (eight occurrences). The effects of stimulation amplitude were not so easy to disentangle (a brief summary of methods used and findings is provided in Appendix A).

### 3.8. Bootstrapping Results

Because of the large number of variables/measures analysed in this study, it was not feasible to bootstrap results for them all. Instead, bootstrapping was only applied to the top two measures for each data type (as listed in Appendix A). As expected, 95% confidence intervals were much narrower (i.e., more precise) for Kendall’s *W* (and Friedman’s *Chi*^2^) for the bootstrapped than the non-bootstrapped results. As an example, in Appendix A the median values of *W* are plotted for differences between time ‘Slots’ at each of the four stimulation frequencies for EEG power at channel P8 in the 1 Hz EEG bin centred on 7.5 Hz (using the extended InfoMax algorithm for ICA and Thomson’s multi-taper method to create the band/bin). For the non-bootstrapped results, the upper confidence interval = 1, so it is not shown. Note that the median *W* is greater for the bootstrapped results (maximal for sham stimulation) and is less for the non-bootstrapped results (maximal for 10 pps). 

### 3.9. Interactions Analysed Using the Hettmansperger–Elmore Test

Using the Hettmansperger–Elmore test, most of the different data types showed interactions between session and stimulation frequency, indicating that despite having allocated stimulation frequencies to sessions in a semi-randomised balanced order, there are still effects specific to combinations of sessions and stimulation frequencies present for unexplained reasons. Counts of *p*-values < 0.01 for the Hettmansperger–Elmore test were transformed into percentages of all counts for each data type (e.g., of 2128 for 1 Hz EEG median power bins, or of 63 for the EEG regional ratios). In general, the percentage counts were lower for the measure differences (values normalised with respect to baseline) than for the values themselves, but all were more than 70%. However, for three data types, significant interactions were found between session and time slot (rather than between session and stimulation frequency): MdnPwr_ASYMM_171 (86.0%), MdnPwr_ASYMM (79.2%), and MdnPwr (78.1%), although these were only for differences, not for values. 

**Results for Objective 2.** Are there meaningful and significant associations between parasympathetic/sympathetic indices and the different eyeblink and EEG-derived data types? 

Several HRV indices are widely accepted as indicating parasympathetic or sympathetic modulation of heart rate [89]. In a 2021 conference poster [25], we proposed extending this list, considering some indices to be ‘PNS-like’, others ‘SNS-like’, and a third group ‘ambivalent’, as shown in Table 2 (based on Tables in the online ‘Background Information’ for the poster). The measures listed are those included in the Kubios HRV software package. 

Spearman correlations between 39 HRV indices and BLINKER measures were then investigated (including those in Table 2 and excluding 20 multiscale entropies in the output from recent versions of the Kubios HRV software). Spearman’s *rho* was computed before, during, and after stimulation at the four different frequencies. BLINKER measures considered were the amplitude–velocity ratios (nAVRZ and pAVRZ), blink rate (BpM), and the ratio of left to right channels providing the best blinks (LRBR). Correlations with the blink rate were consistently positive for the HRV SNS-like measures and negative for the PNS-like measures. Correlations with BLINKER durations, nAVRZ, and pAVRZ were less consistently associated with one or the other, and LRBR appeared almost uniformly PNS-like (except for in the 10 pps group at baseline). The results—excluding durations, durations, nAVRZ, and pAVRZ—are shown in Table 3. 

The following correlations with |*rho*| > 0.3 were also noted: (a) five negative between the standard deviation of good blink duration and various SNS-like measures (SI, HRmin, Hrmean, and SNS itself) and also five negative between ‘goodRatio’ (the ratio of good to all blinks) and various PNS-like measures (RMSSD/SD1, D2, NNxx, and pNNxx), all of these mostly at baseline and preceding sham stimulation; (b) 17 positive, of which 11 were between the standard deviation of good blink duration and PNS-like measures (3 of these at baseline and before sham stimulation and 8 post-stimulation at 2.5 pps). Correlations between the HRV indices and other data type measures are presented in the Appendix A. 

### 3.10. ‘Best’ Blink Laterality and Autonomic Modulation of HRV

For our study participants, many more ‘best’ blinks occurred on the left than on the right (1031 for channel Fp1, 735 for Fp2, 50 for F7, and 19 for F8, with 11 each for F3 and F4). This pattern was found consistently, irrespective of session (1 to 4), stimulation frequency (Sham, 2.5, 10, or 80 pps), and time slot (1 to 8). However, counts of left and right best blinks over time within sessions indicated that the ratio of these counts (LRBR) differed most significantly from the median *during* rather than before or after stimulation (Figure 7). 

Furthermore, counts of left and right best blinks were most different for stimulation at 10 pps, followed by 2.5 pps, and were not significantly different for sham stimulation (Figure 8).

In summary, differences in counts of left and right ‘best’ blinks did not achieve statistical significance at baseline (Slot 1) or at the start of stimulation (Slot 2). Count differences were largest during the middle 10 min of stimulation and then stabilised, with smaller *p*-values. When data were split by stimulation frequency, the binomial test yielded significant results during stimulation *only* for all frequencies, apart from 2.5 pps, for which differences remained significant immediately post-stimulation (Slot 6). As stated by Kleifges et al. (2017) [3], a continuing theme in research on ocular indices is that significant differences are found between individuals, but with consistent patterns in conditions such as perceived sleepiness. Here, we found that significant left blink dominance was found in 14 of our study participants, but significant right blink dominance only in 3 (significance assessed using the binomial test of difference in distribution between left and right blinks). Mann–Whitney tests were also conducted for the 39 Kubios HRV indices, using left or right laterality as the grouping variable. Significant differences for best blink laterality are shown in Table 4. 

Note that left/right laterality patterns were significantly different, with *p* < 0.01 for 18 HRV measures only during stimulation at 2.5 pps. 

Of these 18 measures, 7 could be classified with some confidence as ‘PNS-like’ (plus a further two with less confidence), 3 as ‘SNS-like’, and a further 6 as ‘other’ (Table 5). As described above, the effect size (ES) was calculated for each Mann–Whitney test as |*Z*|/√(*n*), where *Z* = the Z-score and *n* = the number of participants involved. 

Note that values of all three SNS-like measures in this Table were greater for right laterality, whereas of the PNS-like measures, this was the case only for SampEn, one of the two measures with least evidence for its PNS-like classification (the other being Correlation Dimension, D2). Post-stimulation, all frequencies taken together, only PNS was significantly different with laterality (*p* = 0.005, *Z* = −2.801, U = 49,611.5, *N* = 681). The median PNS was larger (−0.624) for left-dominant blink and smaller (−0.805) for right-dominant blink, in keeping with the results in the above table. However, SNS, SI, SD2/SD1, DFA α1, and several frequency-domain measures also differed with laterality exactly as in Table 5, although with less significance (*p* < 0.05 rather than 0.01). Regardless of *p*-values, for these comparisons, the effect sizes shown were all between 0.1 (‘small’) and 0.3 (‘moderate’) [90], a useful reminder that even small *p*-values do not always provide a complete picture. 

### 3.11. Autonomic Associations for the Other Data Types

As noted above (Table 3), absolute values of Spearman’s *rho* for correlations between Kubios HRV and BLINKER measures were mostly between 0.2 and 0.3, for data split into baseline/stimulation/post-stimulation and by stimulation frequency. Very few values of *rho* lay outside that range. In contrast, for the spectral centroids in the ‘centroids’ data type, many values of |*rho*| were > 0.3. When *only* these were considered, the following patterns emerged: 

Positive correlations with HRV measures only occurred for Baevsky’s SNS-like ‘Stress Index’ (SI) [91], while negative correlations were all either with PNS-like measures (HFabs, NNxx, RMSSD, etc.) or with ‘ambivalent’ measures such as SDHR or total power. Furthermore, these correlations occurred only at 6 of the 19 scalp electrodes, Cz, P7, P3, Pz, P4, and P8, with most at those on the midline (Cz and Pz). 

Correlations (median values of |*rho*|) were greatest before stimulation, decreasing thereafter, and were also lower for sham than for the active stimulation frequencies. 

Further analysis of correlations between the HRV indices and data type measures is presented in Appendix A, ‘Autonomic Correlates with Other Data Types’. However, scatter plots did not suggest that the correlations found would be of any great use, practically, as they showed no obvious patterns, even for Spearman’s correlations with *p* < 0.001. 

**Results for Objective 3.** Are measures predictive of participant dropout or adverse reactions to stimulation? 

### 3.12. Dropout

At baseline (Session 1, Slot 1), Mann–Whitney tests indicated that measures of all data types, bar one, including BLINKER and CEPS-BLINKER, as well as questionnaire and scale scores, did not show significant differences between those who subsequently dropped out (6) and those who did not (60) (some reasons for dropping out are given in the next section). In contrast, eight HRV measures at baseline did appear to predict subsequent dropout, including DFA α2 with an effect size of 0.366 (*p* = 0.008), RR, and mean HR with ES 0.349 (*p* = 0.006). Heart rate was—perhaps surprisingly—*lower* at baseline in those who subsequently dropped out than in those who did not. However, the fact that multiple comparisons are being carried out should be considered when interpreting these findings, and caution applied. 

### 3.13. Adverse Reactions

Participants were encouraged to report feeling uncomfortable at any time and were also asked to report any adverse reactions after taking part in the study. Six women and six men did so, whereas fifty participants did not. Physical reactions were mostly minor, including transient nausea, dizziness, or feeling tired, or general discomfort due to the room being too hot. Some participants found skin preparation for the EEG data collection uncomfortable, and nine participants indicated that wearing the EEG cap was itself uncomfortable, resulting in a transient headache for two. At least two reported pins and needles in the hands following electrical stimulation, but this did not last more than a few hours and was not experienced after all four sessions. Three stated that they felt faint towards the end of one session but nonetheless returned for further sessions without further problems. One participant who had previously suffered from a vestibular condition and had experienced a short spell of dizziness in the week prior to the study reported more lasting dizziness following stimulation to the hands and so withdrew from the study following their first session. Another who found ‘the whole situation stressful and unpleasant’ also withdrew after the first session. A third participant withdrew after the first session, without explanation, and a further participant did not attend for their fourth session. Two others also left the study early. Twelve participants were asked, during their sessions, if they could slow their blinking rate, and one attempted to do so without prompting. Eight of the twelve participants reported spontaneously that they found wearing the electrode cap uncomfortable, as did one other. Three participants experienced a headache, and three experienced tiredness following the sessions. One of the latter likened symptoms of general achiness and low mood to those of possible fibromyalgia experienced two years earlier and then dropped out of the study. Three others felt faint (two of them with accompanying nausea) but recovered and continued to take part. 

However, again at baseline, no BLINKER measures predicted subsequent adverse effects, whereas 17 CEPS-BLINKER did so, with *p* < 0.01 and ES between 0.336 and 0.420 (Table 6). In contrast, no questionnaire results predicted adverse effects or dropout with this level of significance. However, a number of other data types, like CEPS-BLINKER, also appeared to predict subsequent adverse effects (Table 7). 

Note that the largest effect sizes (0.420) were both for Hjorth complexity variants of the standard BLINKER indices. 

Note that both maximum and median effect sizes are greater for CEPS-BLINKER than for the other data types, and that the percentage of measures showing significant differences between those that appear to predict subsequent adverse effects and those that do not is also greater for CEPS-BLINKER. 

## 4. Discussion

This study explored the effects of transcutaneous electroacupuncture stimulation (TEAS) on eyeblink rate and other parameters, electroencephalography (EEG), and heart rate variability (HRV). Using a range of non-parametric statistical methods, we analysed physiological responses to different TEAS frequencies, session orders, and time slots within sessions. For the majority of data types analysed, the test statistics and effect sizes were greater when comparing time slots within sessions than when comparing stimulation frequency or session order (see Table 1, Appendix A). As shown in Figure 6, early changes (between Slots 1 and 3) appear to be greater for 80 pps stimulation (in line with reports on high-frequency TENS in the literature [31]), with a cumulative effect for all TEAS frequencies after 20 min of stimulation, although this was most consistently maintained only for 10 pps stimulation thereafter. Our key findings suggest that TEAS exerts significant influences on eyeblink parameters, EEG power spectra, and HRV indices. Notably, eyeblink measures, often disregarded as artefacts in EEG research, emerged as potential indicators of TEAS effects and autonomic modulation. 

One of the novel contributions of this study is the identification of eyeblink parameters as useful physiological markers in acupuncture-related research. Blink rate (BpM) was found to correlate positively with HRV indices associated with sympathetic nervous system (SNS) activity and negatively with parasympathetic nervous system (PNS)-related indices. These correlations reinforce the hypothesis that eyeblink dynamics may reflect autonomic balance, particularly in response to TEAS. Additionally, the laterality of ‘best’ blinks (i.e., those whose trajectory was closest to that of a stereotypical blink) exhibited significant associations with autonomic modulation, particularly during 2.5 Hz stimulation. Right-dominant blink patterns were more strongly correlated with SNS-like measures, whereas left-dominant blink patterns were associated with PNS-like measures. These findings align with previous research suggesting that eyeblink asymmetries may be influenced by neurological or neurochemical states. 

In recent years, a number of researchers have investigated correlations between EEG features and HRV or stress measures [47,48,51]. There is also increasing interest in using HRV itself as a biomarker of altered autonomic function. In one critical review, for example, negative affect (emotion or mood) was found to be associated with a reduction in parasympathetic activity as assessed using HRV [92]. Additionally, there is a growing body of literature on the effects of EA and TEAS on HRV (see, for example, the literature review in [93]). The authors of one study found that low-frequency EA (2.5 Hz) showed a measurable reduction in sympathetic stress with subsequent improvement in vagal tone, whereas medium-frequency EA (15 Hz) elicited changes in all HRV markers, but these failed to achieve statistical significance [94]. In an earlier study, 2 Hz EA appeared to increase vagal activity, whereas 15 Hz EA increased sympathetic activity [95]. Other researchers found a significant increase in the standard deviation of the normal-to-normal heart rate interval (SDNN), usually considered an index of parasympathetic modulation, following 120 Hz EA, with no such change following the 2 Hz EA group [96]. Such findings suggest inconsistencies in the literature that invite further investigation, particularly regarding a central FFR to peripheral stimulation such as TEAS. An incidental finding in the present study was that a more positive mood at baseline might be associated with a decrease in markers of SNS HRV modulation with stimulation (Appendix A). This again suggests further avenues to explore. 

Our results demonstrate that TEAS frequency significantly affects EEG power, particularly within the 1 Hz bins and median power asymmetries, but has less effect on more abstruse measures such as cordance or the Wackermann global descriptors for the 4 sec segments in each 5 min slot (Appendix A). The maximum effect sizes were observed in EEG bands associated with sensory and cognitive processing, particularly at lower stimulation frequencies. This finding may support some prior research indicating that low-frequency electroacupuncture (EA) preferentially enhances vagal activity, whereas higher frequencies may have mixed or less pronounced effects, but provides little evidence in favour of—or against—a central (EEG) FFR in response to TEAS as a form of ‘brain entrainment’. 

HRV analysis revealed that 2.5 Hz TEAS elicited the most significant autonomic changes, primarily increasing vagal tone while reducing sympathetic activity. These results align with other research suggesting that low-frequency stimulation modulates autonomic balance in a way that promotes relaxation and stress reduction [97]. Conversely, higher stimulation frequencies, particularly 80 pps, showed less consistent effects on HRV markers, further highlighting the need for a more nuanced understanding of frequency-dependent autonomic modulation. 

The methodological approach employed in this study was largely exploratory, utilising non-parametric methods such as Friedman’s test, Kendall’s *W*, and bootstrapped Wilcoxon and Conover–Iman tests. Given the non-normal distribution of most measures, these approaches provided a robust means of detecting significant effects while reducing the risk of Type I (false positive) and Type II (false negative) errors. Effect size measures (Kendall’s *W* and Spearman’s *rho*) proved particularly valuable in quantifying physiological changes over time, across TEAS frequencies, and between sessions. However, one challenge in dealing with a large dataset encompassing over 15,000 measures was the need for ‘top slicing’ to identify the most relevant physiological markers. Various slicing methods produced consistent rankings, suggesting that our approach was reliable. Nonetheless, given the multiple comparisons issue, our results should be interpreted with caution, and future studies should consider refining statistical methods and feature selection criteria to enhance reproducibility and interpretability. 

A key objective of this study was to determine whether physiological markers could predict participant dropout or adverse reactions to TEAS. Notably, baseline HRV measures, particularly DFA Alpha2 and mean heart rate, differentiated those who subsequently withdrew from those who completed all sessions. Additionally, CEPS-derived BLINKER measures outperformed standard EEG and HRV indices in predicting adverse effects, suggesting that complexity and entropy analyses of eyeblink parameters may serve as early indicators of discomfort or sensitivity to unfamiliar sensations such as those of electrical stimulation. Furthermore, it is somewhat easier to calculate CEPS-BLINKER measures than those derived from the EEG itself, which in general require time-consuming preprocessing. Indeed, some eyeblink measures could potentially be assessed from video rather than EEG recordings. It may therefore be worth considering the CEPS-BLINKER method of predicting adverse events in other research. 

## 5. Conclusions

Firstly, for the majority of data types analysed, whether derived from eyeblink parameters, EEG measures or HRV indices, test statistics and effect sizes were greater for the effects of time (differences between baseline, stimulation and post-stimulation phases, for example) than for those of stimulation frequency or session order. Secondly, significant autonomic (PNS or SNS) correlates for some eyeblink parameters and EEG measures appear likely. Thirdly, HRV and CEPS-derived BLINKER measures at baseline were early indicators of subsequent adverse effects. 

This study provides novel insights into the physiological effects of TEAS, highlighting the utility of some eyeblink parameters, measures based on EEG power asymmetries, and HRV indices in understanding acupuncture mechanisms. Our findings suggest that TEAS effects are frequency-dependent, with 2.5 Hz stimulation demonstrating the most consistent autonomic modulation. Eyeblink laterality and CEPS-BLINKER measures emerged as promising, though as yet underutilised, biomarkers of TEAS effects. Future research should focus on integrating these markers into broader frameworks of acupuncture efficacy and patient safety monitoring. 

Eyeblink indices could serve as biomarkers for acupuncture effects, with HRV measures at baseline predicting participant dropout and adverse reactions better than EEG-derived indices. These results demonstrate that TEAS modulates brain activity and autonomic function in a frequency-dependent manner and that eyeblink measures, traditionally seen as artefacts, may hold diagnostic and research value. Given their strong correlation with autonomic indices here, further studies should validate their potential as non-invasive biomarkers in acupuncture and other neuromodulation research. 

The differential effects of TEAS frequencies on HRV indices suggest that individualised frequency selection may optimise therapeutic outcomes, particularly for conditions involving autonomic dysregulation. Future research should explore whether TEAS-induced changes in EEG asymmetry correspond to clinical improvements in cognitive and affective disorders. Another avenue to explore is how data length affects different measures. Here, we used 5 min recordings. In parallel studies, time ‘Slots’ 1, 4, and 7 were subdivided into 10 s epochs, resulting in different shortlists of potentially useful measures [29,30], with further reports in preparation. It is anticipated that some measures may provide significant results at both timescales. A next step will be to compare results using the measures from the different shortlists to determine which approach is likely to be the most fruitful, or whether they are complementary.

Our findings suggest that CEPS-enhanced eyeblink analysis could be applied in clinical trials to predict and mitigate adverse responses to electroacupuncture or other neuromodulatory interventions. 

### Innovation and Limitations

The conclusions of this study are derived from analysing a plethora of measures drawn from a relatively small sample of 66 participants. Many of these measures have not been used before in acupuncture-related research, and indeed some data (e.g., that collected from head movement sensors) are only now being analysed. Data were recorded in less-than-perfect circumstances (varying lab temperature, external noise, occasional requests to slow down EBR), and effect sizes were often small, if not very small. Nonetheless, patterns of response were sufficiently uniform (often strikingly so) that we hope this study will encourage other researchers to develop and refine our findings using a larger sample of individuals and a smaller selection of measures derived from eyeblink, EEG, and ECG data.

## Figures and Tables

**Figure 1 sensors-25-04468-f001:**
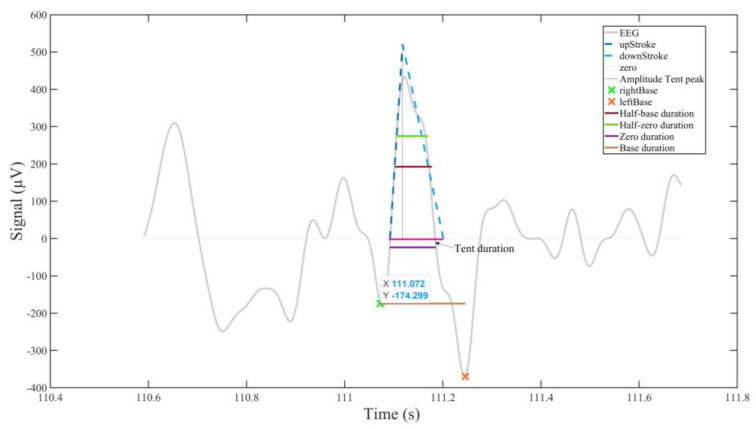
Some eyeblink landmarks and the BLINKER outputs derived from them (based on [44], modified and reproduced with the author’s permission). Details of the ocular indices generated are provided in the Appendix A.

**Figure 2 sensors-25-04468-f002:**
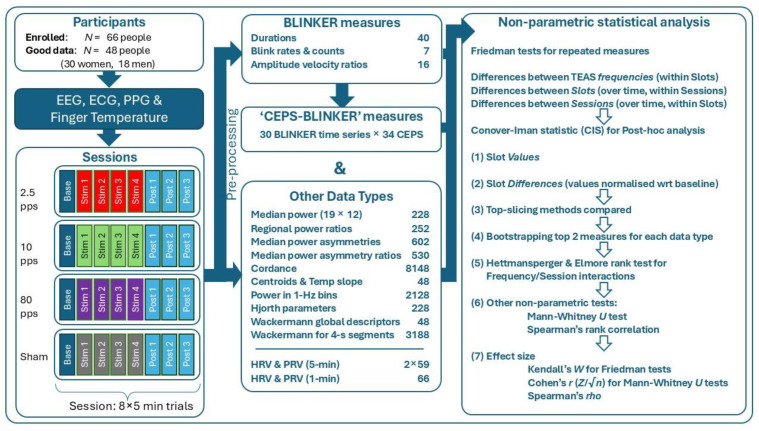
Flowchart of methods used to acquire and analyse EEG, ECG, PPG, and finger temperature data. Background colours indicate the stimulation frequencies applied.

**Figure 3 sensors-25-04468-f003:**
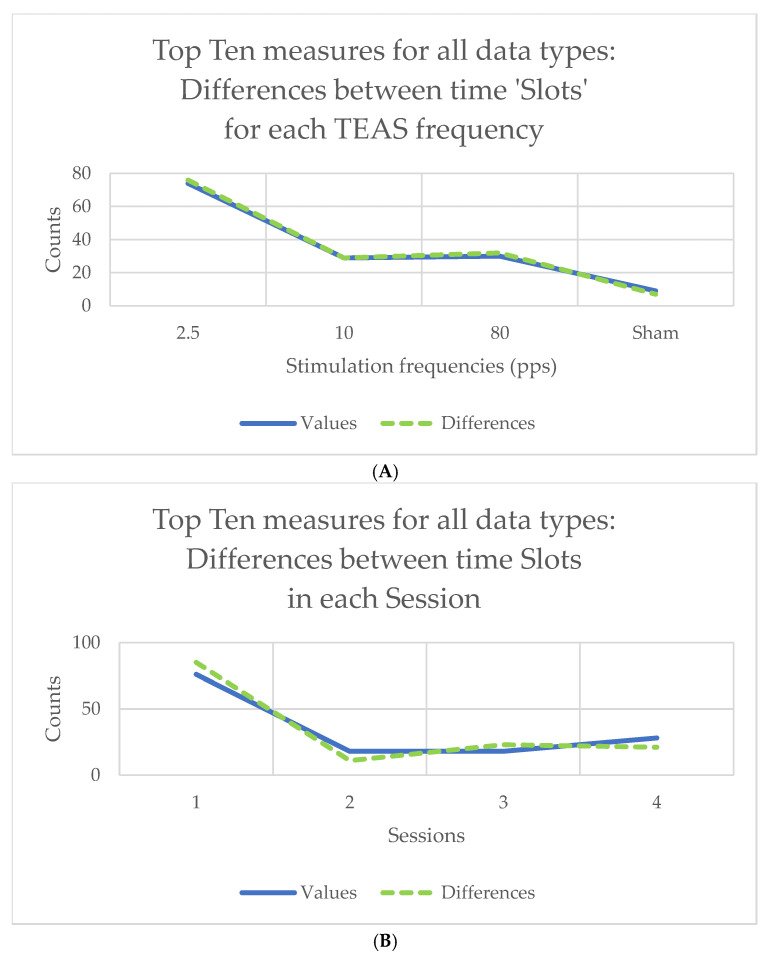
Counts of differences for the 140 ‘top 10’ measures (10 for each of 14 data types): (**A**) between time ‘slots’ for the four stimulation frequencies used; (**B**) between time ‘slots’ in the four sessions attended; (**C**) counts of differences between stimulation frequencies over time, showing changes with session. Counts are shown for both slot values and differences of slot values from baseline, so the latter are zero in (**C**) (see above, Section 2.5, for explanation).

**Figure 4 sensors-25-04468-f004:**
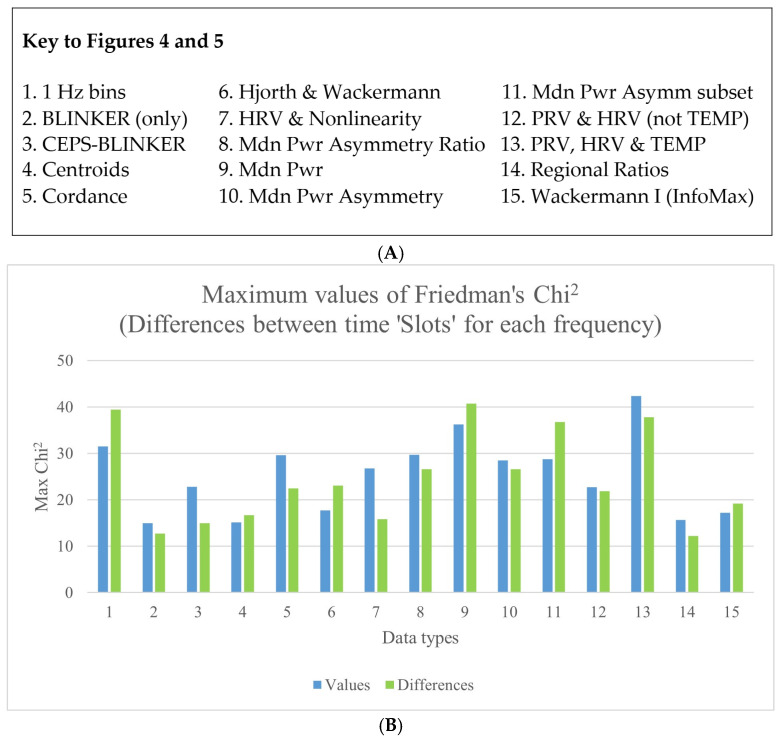
(**A**) A key to explain the numbering system used on the X-axis. (**B**) Differences between stimulation frequencies within each slot, for the various data types—both values and differences.

**Figure 5 sensors-25-04468-f005:**
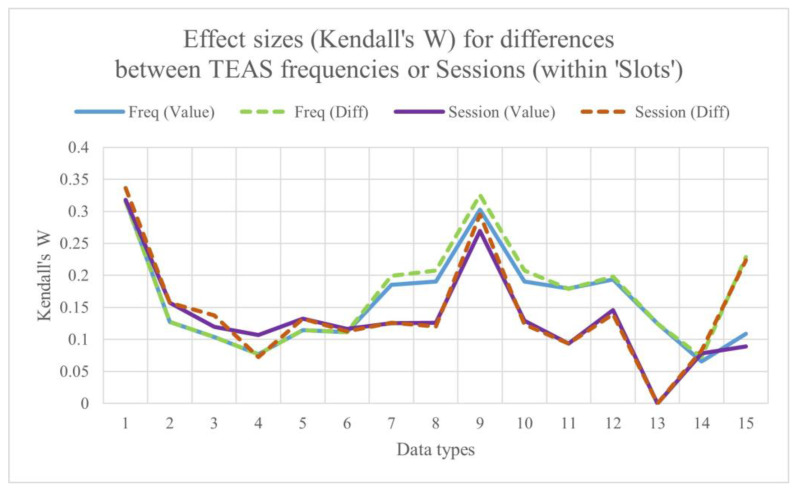
Maximum effect sizes for differences between sessions and differences between stimulation frequencies, both within time ‘slots’.

**Figure 6 sensors-25-04468-f006:**
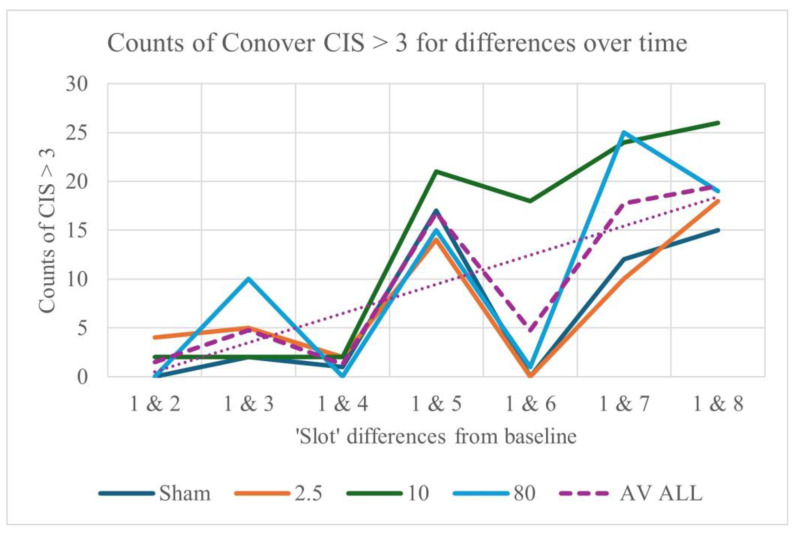
Differences from baseline in counts of CIS > 3 for the different stimulation frequencies, for 63 BLINKER output measures. Median counts for all seven pairs of slots are shown in the inset box. The ‘average’ trendline is dotted.

**Figure 7 sensors-25-04468-f007:**
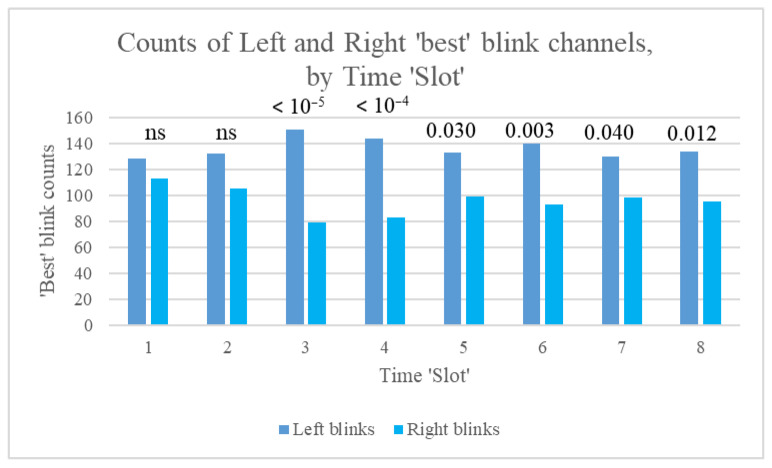
Counts of left and right best blinks over time (Slots 1 to 8), showing *p*-values for the binomial test of difference in distribution between left and right blinks.

**Figure 8 sensors-25-04468-f008:**
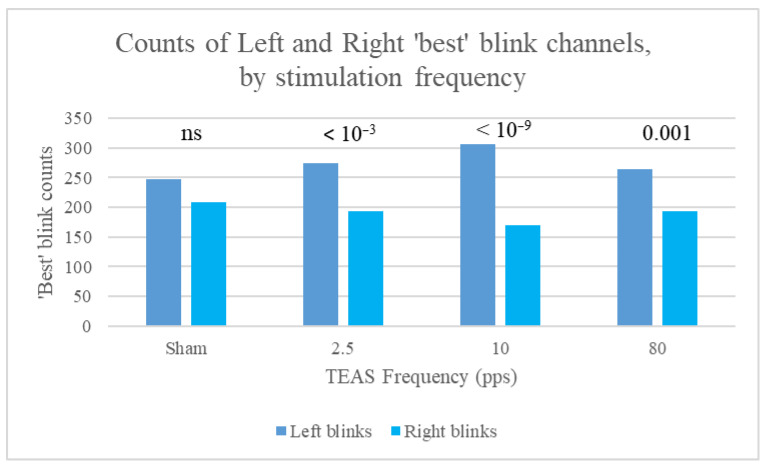
Counts of left and right best blinks for the different stimulation frequencies (all slots taken together), for data split by stimulation frequency, with *p*-values for the binomial test of difference in distribution between left and right blinks. Note the lack of significance for sham stimulation.

**Table 1 sensors-25-04468-t001:** Values of CIS using different ‘top slicing’ methods (maxima, 90th and 95th percentiles, and upper quartiles), for differentiating between time slots at each stimulation frequency (28 comparisons), between frequencies within each time slot, and between sessions within each time slot (6 comparisons each). Median values are also shown.

Method	CIS for Frequencies		CIS for Time Slots		CIS for Sessions	
	BLINKER	CEPS-BLINKER	BLINKER	CEPS-BLINKER	BLINKER	CEPS-BLINKER
Max	3.713	4.445	5.826	5.682	3.894	5.104
95%	2.126	1.968	3.132	2.306	1.963	2.024
90%	1.799	1.650	2.591	1.923	1.664	1.724
Q3	1.222	1.146	1.800	1.346	1.134	1.203
Q2	0.752	0.679	1.055	0.788	0.667	0.726

**Table 2 sensors-25-04468-t002:** ‘Core’ measures considered as indicating parasympathetic (‘PNS-like’) or sympathetic (‘SNS-like’) modulation of heart rate, with a third group of ‘ambivalent’ measures, and others for which this autonomic classification did not seem relevant. The occupational medicine guideline list included as a comparison is from Sammito et al. (2024) [89] and does not consider the PNS, SNS, SI, HFlog and LFlog, SD2/SD1, and EDR (‘electrocardiogram derived respiration’) indices provided by Kubios HRV (so listed as ‘n/a’, not applicable).

Autonomic Classification	HRV Indices	Occupational Med Guideline
PNS-like	PNS	n/a
	SDNN	PNS and SNS
	RMSSD/SD1	PNS
	NNxx	PNS
	pNNxx	PNS
	TI	No clear assignment
	TINN	No clear assignment
	HF%	PNS
	HFabs	PNS
	HFlog	n/a
	HFnu	PNS
	SampEn	No clear assignment
	CorrD (D2)	No clear assignment
SNS-like	SNS	n/a
	SI	n/a
	LF%	PNS and SNS
	LFnu	PNS and SNS
	LF/HF	‘mental workload’
	SD2/SD1	n/a
	DFA α1	No clear assignment
Ambivalent	SDHR	n/a
	TotPwr	No clear assignment
	LFabs	PNS and SNS
	LFlog	n/a
	SD2	PNS and SNS
Other	HF.Hz	No clear assignment
	ApEn	No clear assignment
	EDR	n/a
	DFA α2	No clear assignment

See list of abbreviations below for an explanation of terms.

**Table 3 sensors-25-04468-t003:** Spearman correlations between Kubios HRV and BLINKER measures, with 0.2 < Spearman’s |*rho*| < 0.3 (if |*rho*| ≥ 0.3, as for several correlations with BpM, this is indicated). HRV indices in red are considered to be ‘SNS-like’, those in blue ‘PNS-like’, and those in italic type as ‘other’.

Pre-Stim-Post	pps	BLINKER Measures	Negative *rho*	Positive *rho*
Pre	Sham	n/a	n/a	n/a
	2.5	BpM	n/a	EDR
	10	LRBR		LF%
	80	LRBR	n/a	SDNN, TINN, TI, PNS, RMSSD, SD1, HFAbs, HFlog
Stim	Sham	BpM	HFabs and log pwr, SD1 and RMSSD, NNxx, pNNxx, D2, *ApEn* and SampEn [<−0.3]	LF/HF, LFnu, SD2/SD1, DFA α1, SNS
	2.5	BpM LRBR	n/a	SNS, SD2/SD1, DFA α1 RMSSD, SD1, HFabs, HFlog
	10	BpM LRBR	n/a	LF/HF, LFnu, DFA α1 LFAbs, LFlog
	80	LRBR	n/a	PNS, NNxx, pNNxx, HFabs, HFlog, Totpwr, RMSSD, SD1, TINN, *SD2*
Post	Sham	BpM LRBR	SD1, RMSSD and pNNxx [<−0.3] PNS, HFabs and HFlog, D2, TI and PNS	SNS, SD2/SD1, DFA α1 PNS (>0.2)
	2.5	n/a	n/a	n/a
	10	BpM	SD1, RMSSD, HF abs and HF log, NNxx, pNNxx, TI, HF% and HFnu; and PNS (<−0.3); D2	SD2/SD1, LFnu, LF/HF, DFA α1, *DFA α2*; SNS (>0.3)
	80	BpM LRBR	NNx [<−0.03]	SNS, SD2/SD1, *DFA α2* (>0.2) PNS, SDNN, TI, SD1, RMSSD and TINN, *SD2*, SDHR

See list of abbreviations below for an explanation of terms.

**Table 4 sensors-25-04468-t004:** Counts of significant differences in 39 HRV indices between left (‘1’) and right (‘0’) laterality groups before, during, and after stimulation at the four different frequencies.

L1 v. R 0	Baseline		Stimulation		Post-Stimulation	
	*p* < 0.05	*p* < 0.01	*p* < 0.05	*p* < 0.01	*p* < 0.05	*p* < 0.01
Sham	12	0	2	0	2	0
2.5	0	0	25	18	3	0
10	0	0	0	0	6	0
80	0	0	1	0	7	0

**Table 5 sensors-25-04468-t005:** The eighteen measures for which left/right laterality patterns were significantly different, with *p* < 0.01 during stimulation at 2.5 pps. R > L indicates those measures which were greater for right laterality, and L > R those which were greater for left laterality. See list of abbreviations and our 2021 conference presentation [25] for further information.

Measures	Classification	*p*-Values	ES	L > R or R > L
SNS	SNS-like	0.006	0.179	R > L
SI	SNS-like	<10^−4^	0.261	R > L
SDNN	PNS-like	<10^−3^	0.232	L > R
HRmin	SNS-like	0.003	0.192	R > L
RMSSD	PNS-like	0.006	0.182	L > R
NNxx	PNS-like	0.007	0.176	L > R
pNNxx	PNS-like	0.008	0.174	L > R
TI	PNS-like	0.008	0.175	L > R
TINN	PNS-like	<10^−3^	0.248	L > R
LFabs	Other	<10^−3^	0.241	L > R
LFlog	Other	<10^−3^	0.241	L > R
TotPwr	Other	<10^−3^	0.235	L > R
SD1	PNS-like	0.006	0.182	L > R
SD2	Other	<10^−3^	0.230	L > R
ApEn	Other	0.001	0.220	R > L
SampEn	PNS-like	0.009	0.172	R > L
DFA α2	Other	0.005	0.185	R > L
D2	PNS-like	0.001	0.219	L > R

**Table 6 sensors-25-04468-t006:** CEPS-BLINKER measures that appear to predict subsequent adverse effects, using Mann–Whitney tests with *p* < 0.01. Median values of the measures and their interquartile ranges are shown in the final two columns, with the greater value for each measure in bold.

CEPS-BLINKER Measure	*p*-Value < 0.01	ES	Adverse Effects	No Adverse Effects
Amplitude_FD_AvRVA	0.004	0.372	1.168	**1.228** (1.193,1.270)
CCM_AvRVA	0.008	0.339	0.222	**0.289** (0.232, 0.348)
DFA_MATS_dHB	0.005	0.355	**0.714**	0.462 (0.233, 0.695)
DFA_MATS_dHZ	0.004	0.370	**0.776**	0.538 (0.242, 0.739)
FD_Linden_Box_AvRVA	0.007	0.342	1.662	**1.758** (1.670, 1.820)
HjorthC_cTT	0.007	0.346	**1.258**	1.180 (1.126, 1.253)
HjorthC_cTZ	0.004	0.365	**1.269**	1.189 (1.1–7, 1.235)
HjorthC_dB	0.003	0.375	**1.287**	1.183 (1.128, 1.263)
HjorthC_dHB	0.005	0.360	**1.312**	1.220 (1.125, 1.290)
HjorthC_dHZ	0.008	0.339	**1.345**	1.221 (1.133, 1.310)
HjorthC_dT	0.002	0.394	**1.385**	1.217 (1.119, 1.299)
HjorthC_dZ	0.001	0.420	**1.408**	1.205 (1.128, 1.287)
HjorthC_AvRVA	0.001	0.420	**1.429**	1.270 (1.179, 1.333)
HjorthM_dT	0.009	0.336	1.134	**1.366** (1.273, 1.496)
HjorthM_dZ	0.003	0.387	1.147	**1.393** (1.235, 1.505)
HjorthM_AvRVA	0.003	0.375	1.157	**1.359** (1.215, 1.424)
Hurst_H_AvRVA	0.006	0.353	0.761	0.564 (0.466, 0.717)

Key: Measures in this table are named according to the CEPS measure (e.g., ‘Amplitude_FD’) applied to a selected BLINKER time series (e.g., ‘AvRVA’). The abbreviations used are listed in the Appendix A.

**Table 7 sensors-25-04468-t007:** Other data type measures that appear to predict subsequent adverse effects, using Mann–Whitney tests with *p* < 0.01. Median values of the measures and their interquartile ranges are shown in the final two columns, with the greater absolute value for each measure in bold.

Data Type	*p*-Value < 0.01	ES	Adverse Effects	No Adverse Effects
Cordance	(6 of 8148)	(<1%)		
CorN_Beta_T5Inf_mT_s	0.003	0.378	**−0.326** (−0.525, −0.235)	0.245 (−0.062, 0.427)
CorN_Alpha_P3Inf_mTR_s	0.007	0.342	**0.465** (0.446, 0.712)	0.268 (0.131, 0.363)
CorN_Beta_T5Inf_mTL_ln	0.003	0.382	**−0.312** (−0.496, −0.214)	0.257 (0.102, 0.424)
CorN_Alpha_P3Inf_mTR_ln	0.008	0.335	**0.471** (0.451, 0.652	0.275 (0.146, 0.368)
SQzTInf_mT_Theta4_8_P4	0.005	0.360	**−1.930** (−2490, −1.533)	−0.543 (−1.086, 0.100)
LNzTInf_mT_Theta4_8_P4	0.005	0.360	**−1.931** (−2.520, −1.621)	−0.505 (−1.088, 0.157)
Hjorth parameters	(1 of 228)	(<1%)		
HjorthM_P3	0.009	0.263	0.139 (0.117, 0.151)	**0.167** (0.153, 0.191)
Wackermann descriptors	(11 of 3188)	(<1%)		
Omega_21_I	0.008	0.339	7.354 (6.500, 8.397)	**9.748** (8.669, 10.631)
Omega_23_A	0.006	0.351	7.622 (4.820, 8.510)	**10.305** (9.115, 10.855)
Omega_26_A	0.003	0.384	6.388 (4.820, 7.944)	**10.155** (9.127, 10.948)
Omega_36_A	0.002	0.392	6.064 (4.022, 8.212)	**10.162** (9.036, 11.015)
Omega_71_A	0.004	0.377	6.240 (4.843, 7.803)	**10.085** (8.878, 10.940)
Phi_26_I	0.005	0.358	11.677 (9.441, 13.356)	**16.498** (14.187, 18.847)
Phi_29_I	0.007	0.347	12.150 (10.382, 12.818)	**16.763** (14.366, 18.534)
Phi_30_I	0.006	0.351	11.682 (10.458, 13.560)	**16.118** (14.036, 18.516)
Phi_73_I	0.008	0.351	11.586 (10.965, 13.243)	**16.364** (14.117, 18.824)
Phi_26_A	0.004	0.369	10.543 (8.684, 13.206)	**16.159** (14.079, 18.307)
Phi_36_A	0.005	0.362	10.085 (6.447, 12.873)	**16.379** (13.741, 18.413)
1 Hz bins	(1 of 2128)	(<1%)		
P8_10.0_A_PS_mT	0.008	0.335	**44.490** (43.424, 50.817)	38.729 (34.363, 42.168)
Asymmetry	(1 of 602)	(<1%)		
R_L_O80 [4,58]	0.008	0.335	**3.305** (1.322, 5.363)	−0.844 (−2.524, 0.829)
Median (IQR)	0.006 (0.004, 0.008)	0.355 (0.341, 0.371)		Max ES 0.392
CEPS-BLINKER	(17 of 1020)	(1.67%)		
Median (IQR)	0.004 (0.003, 0.007)	0.365 (0.346, 0.375)		Max ES 0.420

## Data Availability

The raw data supporting the conclusions of this article will be made available by the corresponding author (DB, email duncan.banks@open.ac.uk), upon reasonable request.

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
