# Peer review of "Effects of Transcutaneous Electroacupuncture Stimulation (TEAS) on Eyeblink, EEG, and Heart Rate Variability (HRV): A Non-Parametric Statistical Study Investigating the Potential of TEAS to Modulate Physiological Markers"

_sensors, 2025, doi:10.3390/s25144468_

Round 1
Reviewer 1 Report
Comments and Suggestions for Authors
This study makes a valuable contribution to neuromodulation and acupuncture research by exploring eyeblink parameters as physiological markers in TEAS.
The variable analysis is thorough, though focusing on the most relevant findings could improve clinical impact. Future presentations should more clearly explain how eyeblink parameters can signal early adverse reactions in both clinical and research settings.
The authors appropriately note limitations, such as the small sample size and the study’s exploratory nature - both important for interpreting results and guiding future research. Overall, the methodology is solid and the presentation clear, providing a strong basis for further work in TEAS-related physiological monitoring.
Author Response
Thank you for your helpful review.
The variable analysis is thorough, though focusing on the most relevant findings could improve clinical impact. Future presentations should more clearly explain how eyeblink parameters can signal early adverse reactions in both clinical and research settings.
Thank you for your comments on how future presentations should explain how eyeblink parameters can signal early adverse reactions in both clinical and research settings. We agree with this comment and will consider it in future publications.
The most relevant findings are summarised in the Discussion and Conclusions, particularly the first paragraph of the latter.
In our view, it is clear (original lines 915-20, now in red) that certain HRV and CEPS-derived BLINKER measures at baseline may be predictive of subsequent drop-out or adverse reactions or readers may wish to adopt the methods described in the paper for application in other clinical and research settings.
Reviewer 2 Report
Comments and Suggestions for Authors
The authors investigated the effects of Transcutaneous Electrical Acupuncture Stimulation (TEAS) on a series of physiological manifestations (including eyeblink parameters, electroencephalogram [EEG] signals, and heart rate variability [HRV], etc.). A suite of statistical analysis methods—including Friedman’s test, the Wilcoxon signed-rank test, the Conover-Iman test, and bootstrapping—were employed to explore the associations between TEAS and these physiological indices, with a particular focus on eyeblink-related parameters. The authors described the experimental design and data analysis protocols in detail throughout the manuscript. The conclusions of this study propose potential biomarkers for evaluating TEAS efficacy.
Here are several questions that require clarification from the authors:
- The technical means of image capture and recognition are highly mature. If blink details can be captured by cameras and analyzed using artificial visual technology, the results may be more accurate than those from biopotential signals.
- Since the number of valid subjects was 48, the authors should not have stated "66" in the abstract.
- The authors are advised to include a diagram in the supplementary materials to clarify how stimulation and physiological signals were applied to the subjects.
Author Response
Thank you for your helpful observations on our paper.
- The technical means of image capture and recognition are highly mature. If blink details can be captured by cameras and analyzed using artificial visual technology, the results may be more accurate than those from biopotential signals.
We agree that technical means of image capture and recognition may be highly mature now (2025), but we designed this study following a series of pilot investigations conducted from 2011 onwards, when image capture was less advanced and EEG was the technology available to us.
We do acknowledge (originally lines 923-4, now in red) that ‘some eyeblink measures could potentially be assessed from video rather than EEG recordings’ (we have not explored that further). We suggest that this could be amended to: ‘some eyeblink measures could potentially be assessed from video rather than EEG recordings using low-cost web cameras on a laptop or tablet, with artificial intelligence to analyse eyeblink and other eye movements and thus could be a useful tool for clinicians and therapists to monitor clients in real time’.
- Since the number of valid subjects was 48, the authors should not have stated "66" in the abstract.
The number initially enrolled was, indeed, 66, but complete Eyeblink, HRV and EEG datasets were, as you observe, only available for 48. Thank you for pointing out this discrepancy. The text should now be amended. Lines 32-3 (‘Sixty-six participants underwent four TEAS sessions’) should now read: ‘Sixty-six participants were enrolled, for whom complete Eyeblink, HRV and EEG datasets were available for 48. These underwent four TEAS sessions …’
- The authors are advised to include a diagram in the supplementary materials to clarify how stimulation and physiological signals were applied to the subjects.
We have already clarified this in previous open-access publications [28-30], as mentioned in Line 142. An appropriate illustration of the TEAS and ECG electrodes, as well as PPG (and temperature) sensors, is provided in all three references. A head map illustrating the approximate location of the EEG electrodes is included in [29]. We hope you agree that it is not necessary to repeat these Figures here, as readers who require the information they contain can easily access them online.
Reviewer 3 Report
Comments and Suggestions for Authors
The authors present their findings on how transcutaneous electroacupuncture stimulation affects physiological measures such as eyeblink rate, heart rate variability, and EEG features.
Here is some feedback I have.
1) There is too much content. The paper is too wordy, and would benefit if it focused down its scope to either eyeblink, HRV, or EEG features.
2) There's a dearth of information on what these findings might mean in a clinical or therapeutic context. This should be explored more clearly in the conclusion.
3) The paper suggests that electrical stimulation of specific acupuncture points causes changes in physiological measures such as eyeblink, HRV, and EEG. However, from what I understand, even in the sham condition, an electrode was placed at the acupuncture point, and stimulated for a while before the amplitude was reduced. This is still electrically stimulating the acupuncture point. Why not instead place the electrode somewhere that's not an acupuncture point at all, and ideally with the participant unable to see where exactly the electrode is?
4) In the same vein as point 1, there are far too many statistical tests being conducted, yet there is no correction for multiple comparisons. It's stated in the text: "However, the fact that multiple comparisons are being carried out should be considered when interpreting these findings, and caution applied." You should do this, e.g. by performing a Bonferroni or Sidak correction on all your p-values. Otherwise, you may be just picking up statistical noise.
5) What exactly are 'top 10' measures? Are you extracting multiple features from each physiological data type, and counting how many features are statistically significant? This section isn't clear.
6) "In total, some 15,356 measures were available for analysis." Do you mean 15,356 features were extracted and then statistically tested? This is analysing far too many features in a single study. Once you perform Bonferroni correction with that, most of the p-values will be statistically significant. I have trouble believing that the data presented is any more than random noise. For example, in Table 7, 6 out of 8148 Cordance features are shown to be statistically insignificant. Apply a Bonferroni correction to these p-values, and the corrected p-values will all be greater than 1! In the first place, it is bad scientific practice to go on a "fishing trip" with no clear a priori hypothesis, because it's possible to find spurious correlations that might appear statistically significant but don't really mean anything. Secondly, once appropriate corrections are made for multiple comparisons, this paper doesn't even seem to have found statistical significance!
Author Response
Thank you for your detailed review of our paper. We will try and address each point in turn.
The following ‘must be improved’:
Does the introduction provide sufficient background and include all relevant references?
We believe that sufficient background is provided, and that relevant references have been included. However, research moves quickly, and because of some health issues experienced by the lead author, it has taken more time than initially anticipated to complete this paper, so that all recent relevant references may not have been included.
Are the methods adequately described? [‘can be improved’]
This reviewer already considers that we have included ‘too much content’ and have been ‘too wordy’. We have described the methods used – whether experimental or analytical – as succinctly as we could while providing all requisite information.
The reviewer does not mention any specific areas where the methods are not sufficiently described so, in the interests of brevity, we are not adding more text (although we would be happy to do so for any specific areas if requested). We include more detail in the supplementary material.
Is the research design appropriate?
We acknowledge that the study explores an unusually large number of variables. For the data collection phase of the study, experienced clinicians acknowledged as leaders in their fields contributed to the research design (TW on electrotherapy and TENS, DM on electroacupuncture and TEAS). Another co-author (NS), Professor of Applied Statistics at the University of Hertfordshire, who has a particular interest in methods of analysing complex data (and a corresponding publishing history) advised on the statistical methods to be used. TS has many years of experience in HRV and EEG data collection and analysis. The research design was discussed extensively and agreed by all authors.
Are the results clearly presented?
Results are summarised in graphical and tabular form throughout the paper and Supplementary Material, as well as in the text.
Are the conclusions supported by the results?
In our view, they are. If there are some conclusions that the reviewer does not find to be supported by the results, please can s/he indicate to us which these are.
Are all figures and tables clear and well-presented?
If there are particular Figures and Tables that could be improved, we would appreciate it if the reviewer could indicate which require attention.
- There is too much content. The paper is too wordy, and would benefit if it focused down its scope to either eyeblink, HRV, or EEG features.
1.1 Too much content; too wordy.
We acknowledge that our paper is quite lengthy, and that we have tried to cover a lot of ground. This has been deliberate, in that two of the authors are now in their 70s and coming to the end of their academic careers, so are keen to publish these findings before they do so. Most of us are established authors with long experience in academic writing, and do not consider our writing style to be unduly ‘wordy’. Furthermore, MDPI state explicitly that “Sensors has no restrictions on the maximum length of manuscripts, provided that the text is concise and comprehensive’, and we feel we have fulfilled these requirements.
1.2. Would benefit if it focused down its scope to either eyeblink, HRV, or EEG features.
The first two of our three study objectives depend on comparisons between the three main data types considered (Eyeblink, HRV, EEG). To focus on a single data type would invalidate the overall structure of our study, and we would certainly prefer to present all these results together, rather than splitting the paper between separate publications.
- There's a dearth of information on what these findings might mean in a clinical or therapeutic context. This should be explored more clearly in the conclusion.
At the risk of burdening the reader further, we could add a sentence to the Conclusions (at the end of the first paragraph): Perhaps ‘These all have potentially useful clinical applications – for example, in the exploration of duration of stimulation, its autonomic and any possible adverse effects’.
- The paper suggests that electrical stimulation of specific acupuncture points causes changes in physiological measures such as Eyeblink, HRV, and EEG. However, from what I understand, even in the sham condition, an electrode was placed at the acupuncture point, and stimulated for a while before the amplitude was reduced. This is still electrically stimulating the acupuncture point. Why not instead place the electrode somewhere that's not an acupuncture point at all, and ideally with the participant unable to see where exactly the electrode is?
This is a sensible suggestion for further study. For example, Zeng et al., in a 2006 paper titled ‘Electroacupuncture modulates cortical activities evoked by noxious somatosensory stimulations in human’ (doi:10.1016/j.brainres.2006.03.123) wrote that ‘A specific later-latency somatosensory-evoked potential (SEP, P150) located in bilateral anterior cingulated cortex was observed after EA acupoint [LI4] but not non-acupoint’.
No single study can answer all possible research questions, and in ours we explicitly stated that our interest was in comparing the effects of specific stimulation frequencies on the physiology of healthy individuals. Our intention was deliberately not to include control points as well, but to focus on the effects of different stimulation frequencies, rather than opening the Pandora’s box of ‘what is an acupuncture point?’ Furthermore, if in three of the four sessions for each participant the electrode was at one location, but at a different location in the fourth, participants would quickly become aware which was the sham intervention.
Stimulation for ‘a while’ was only for a matter of seconds, no longer (‘once it had been felt initially by the participants’, line 158), and at a very low amplitude that would be considered therapeutically irrelevant by most clinicians.
The reviewer’s suggestion of positioning the electrode/s where participants cannot see them is certainly attractive, but once stimulation was started (whether verum or sham), it would be clear to participants where the electrodes were positioned, and in our view the sensation elicited would be more important to the participant than whether they can see the electrode/s or not. It was also important that the electrode/s could be seen by the experimenter in case of dislodgement.
- In the same vein as point 1, there are far too many statistical tests being conducted, yet there is no correction for multiple comparisons. It's stated in the text: "However, the fact that multiple comparisons are being carried out should be considered when interpreting these findings, and caution applied." You should do this, e.g. by performing a Bonferroni or Sidak correction on all your p-values. Otherwise, you may be just picking up statistical noise.
The reviewer is correct that ideally adjustments to the p-values would be made so as to allow for the fact that multiple comparisons are being undertaken. However, the Bonferroni and Šidák corrections suggested are only appropriate for situations where the multiple tests are independent of each other and this is far from being the case in this paper. Other methods often employed to adjust p-values also assume the tests are independent of each other. For situations, such as occur in this paper, where the multiple tests are not independent, simulation methods can, in theory, be used to make adjustments (see, for example, a paper by one of the authors of this current paper: Spencer, N.H. (2009) “Overcoming the multiple testing problem when testing randomness”. Journal of the Royal Statistical Society, Series C, 58, 4, pp543-553). However, in the current context, with a very large number of highly correlated tests, it is not computationally feasible to undertake such simulations. As a result, it is most appropriate to take the p-values as they are and apply appropriate scientific judgment and caution when drawing conclusions.
- What exactly are 'top 10' measures? Are you extracting multiple features from each physiological data type, and counting how many features are statistically significant? This section isn't clear.
We thank the reviewer for pointing this out, and will modify the paper as follows:
Line 447 (in the original, now in red): ‘pragmatic way of proceeding’ becomes: ‘pragmatic way of proceeding, i.e., taking the top 10 values of Friedman’s Chi-Square or Kendall’s W for all cases for a particular comparison – between Slots or frequencies, for example.’
- "In total, some 15,356 measures were available for analysis." Do you mean 15,356 features were extracted and then statistically tested? This is analysing far too many features in a single study. Once you perform Bonferroni correction with that, most of the p-values will be statistically significant. I have trouble believing that the data presented is any more than random noise. For example, in Table 7, 6 out of 8148 Cordance features are shown to be statistically insignificant. Apply a Bonferroni correction to these p-values, and the corrected p-values will all be greater than 1! In the first place, it is bad scientific practice to go on a "fishing trip" with no clear a priori hypothesis, because it's possible to find spurious correlations that might appear statistically significant but don't really mean anything. Secondly, once appropriate corrections are made for multiple comparisons, this paper doesn't even seem to have found statistical significance!
A large number of tests have been carried out but, rather than this being because we were carrying out a trawling expedition to harvest small p-values, it was because with the current level of scientific knowledge, it was not possible to hypothesise how patterns of interest would be revealed. An approach which restricted our explorations to just a small number of tests would risk making a priori judgements about the underlying scientific processes. However, we acknowledge that the presence of a large number of tests presents the challenge of how to assess statistical significance. For reasons explained above, it is not possible to apply adjustments to p-values and other methods of assessing importance must be applied. We have done this by looking at patterns where multiple tests are pointing in the same direction – something that is unlikely to occur if the data is simply random noise. Our approach is one that was the subject of a presentation at the 2022 Conference of the Royal Statistical Society by one of the authors of this paper which received very positive feedback (Spencer, N.H., Mayor, D., Steffert, A. & Beggan, C.D. (2022) “From a Berkshire Farm to a Fishing Expedition via Random Points: A Statistician’s Journey”. Conference of the Royal Statistical Society, Aberdeen, UK, September 2022) [81].
If the reviewer agrees we suggest adding the following to the limitations section of the paper "Our top-slicing approach selects only the most significant findings. This increases the likelihood of identifying effects that achieve significance by chance alone, so they should be interpreted with caution. Nonetheless, we believe our study can provide a useful starting point for further research, and we encourage others to validate our results in independent datasets."
We believe the reviewer has made a typo, and that in Table 7, 6 out of 8148 Cordance features are shown to be statistically significant. As has already been pointed out, it is not appropriate to use a Bonferroni correction for non-independent tests. In support of our view that the results we have found are not spurious, we would like to repeat that, in our view, p-values are not the only method of analysis that can be used. As we state repeatedly throughout our paper, patterns of change or difference were frequently consistent (often strikingly so), regardless of p-values.
Thank you for being so thorough in your review. Our data were, indeed, highly complex and required the use of multiple comparisons. Furthermore, our experiment was conducted without any preconception of a particular result, which is why we worded our objectives as research questions rather than hypotheses.
Round 2
Reviewer 3 Report
Comments and Suggestions for Authors
Thank you for the response. I think the other points have been adequately addressed, however:
"The reviewer is correct that ideally adjustments to the p-values would be made so as to allow for the fact that multiple comparisons are being undertaken. However, the Bonferroni and Šidák corrections suggested are only appropriate for situations where the multiple tests are independent of each other and this is far from being the case in this paper. Other methods often employed to adjust p-values also assume the tests are independent of each other. For situations, such as occur in this paper, where the multiple tests are not independent, simulation methods can, in theory, be used to make adjustments (see, for example, a paper by one of the authors of this current paper: Spencer, N.H. (2009) “Overcoming the multiple testing problem when testing randomness”. Journal of the Royal Statistical Society, Series C, 58, 4, pp543-553). However, in the current context, with a very large number of highly correlated tests, it is not computationally feasible to undertake such simulations. As a result, it is most appropriate to take the p-values as they are and apply appropriate scientific judgment and caution when drawing conclusions."
You're essentially saying, p-value adjustments to address the multiple correction problem are not necessary if you use simulation methods instead... but you didn't perform these simulation methods either because it's not computationally feasible to do so. So you want to publish your conclusions while using neither standard p-value adjustments nor simulation methods. This is not scientifically sound.
Author Response
Final feedback please see response to academic editor.